# Late Luteal Subphase Food Craving Is Enhanced in Women with Obesity and Premenstrual Dysphoric Disorder (PMDD)

**DOI:** 10.3390/nu15235000

**Published:** 2023-12-02

**Authors:** Ajna Hamidovic, Shahd Smadi, John Davis

**Affiliations:** 1Department of Pharmacy, College of Pharmacy, The University of Illinois at Chicago, Chicago, IL 60612, USA; 2Department of Psychiatry, College of Medicine, The University of Illinois at Chicago, Chicago, IL 60612, USA; ssmadi2@uic.edu (S.S.); davisjm@uic.edu (J.D.)

**Keywords:** appetite, food intake, PMDD, abdominal obesity, CRP

## Abstract

Dysregulated food craving is a complex weight-related behavior. To identify novel targets for enhancing the efficacy of weight loss interventions, we examined whether food craving varies across the menstrual cycle according to the abdominal obesity type and premenstrual dysphoric disorder (PMDD) diagnosis, and, if so, whether it is related to affective symptomatology. Reproductive-age women were classified into one of the four study groups according to whether they have abdominal obesity (AO) or are abdominally lean (AL), and the presence of PMDD: (1) AO:PMDD+ (n = 13), (2) AL:PMDD+ (n = 14), (3) AO:PMDD− (n = 15), and (4) AL:PMDD− (n = 16). Self-report measures as well as urinary luteinizing hormone (LH) tests were provided by the participants across 2–3 menstrual cycles. The ratings of food cravings were similar across the menstrual cycle, except the last, late luteal subphase as the AO:PMDD+ participants had the highest food craving rating. Irritability and depression were correlated with food cravings, but not in a distinctive manner across the menstrual cycle by group. Our study found that women with abdominal obesity and PMDD display a temporal vulnerability to a food-related behavior. The possibility of shared neurobiology between the two conditions is discussed and should be examined in future studies.

## 1. Introduction

Premenstrual dysphoric disorder (PMDD) is a psychiatric condition affecting between 3 and 5% of reproductive-age women [1] who experience debilitating affective symptomatology in the luteal phase of the menstrual cycle. Symptom expression in PMDD is, among other factors, related to body fat accumulation, as childhood (pre-menarche) body mass index (BMI) is positively associated with the development of PMDD in adulthood [2], and women with obesity have an increased risk for PMDD [3]. 

The menstrual cycle may represent a trigger for the obesity and PMDD co-expression. The luteal phase of the menstrual cycle is marked by a natural mild anabolic state across reproductive-age women [4] and a small parallel increase in energy intake [5] to maintain energy balance. This luteal phase equilibrium, however, is distorted in women with PMDD as they report higher food cravings than the general population of reproductive-age women [6] who also report this increase (relative to the follicular phase of the menstrual cycle). Premenstrual food cravings are likely influenced by several factors, not only the interacting effects between abdominal obesity and PMDD status, as hypothesized here. For example, we previously [7] investigated hormonal influences on premenstrual food cravings using hierarchical linear regression modeling, showing that progesterone, but not estradiol, is inversely associated with premenstrual food cravings.

It is presently not known whether the enhanced premenstrual food cravings in women with PMDD is related to abdominal obesity. Hence, we prospectively evaluated food cravings across the entire menstrual cycle in relation to abdominal obesity and PMDD diagnosis. We next assessed if group and subphase effects on food cravings may be associated with depression and/or irritability across the menstrual cycle. We also collected high-sensitivity c-reactive protein (hs-crp) in a subset of women and preliminarily examined the relationship between hs-crp and food cravings in the late luteal subphase of the menstrual cycle in PMDD. We deemed this analysis necessary as the inflammatory protein is a marker of poor dietary intake [8], and it is unknown whether premenstrual crp is associated with PMDD.

## 2. Materials and Method

### 2.1. Study Design

The present analysis is a part of the Premenstrual Hormonal and Affective State Evaluation (PHASE) project that is described in detail in Hamidovic et al. (2022) [9]. In summary, study participants were reproductive-age women with PMDD and healthy controls free of other current psychiatric conditions. They were non-smokers, did not take any prescription medications, including hormonal forms of birth control, and did not take illicit drugs as verified by a urine drug test. They also did not have other current psychiatric co-morbidities. The participants provided symptom ratings using the Daily Record of Severity of Problems (DRSP) [10] across two to three menstrual cycles. They completed these ratings between 8 PM and midnight on a daily basis. In the last menstrual cycle of the study, a subset of women from PHASE attended 8 clinic visits throughout the menstrual cycle to provide serum samples, complete tests of cognition and measures of heart rate variability. We analyzed hs-crp levels from the mid-follicular and late luteal subphase visits, which we present here.

### 2.2. Food Cravings and Anthropometric Measures

Food Cravings Measure. The Daily Record of Severity of Problems (DRSP) [10] is a validated instrument that measures affective, psychological, behavioral, and functional symptoms. They are rated on a scale of 1 (not at all) to 6 (extreme). The internal consistency and test–retest variability of the instrument are high [10]. The present analysis evaluated ratings of increased food cravings (“Had cravings for specific foods”). This daily rating of food cravings was calculated as the mean score for the following timepoints: (1) mid-follicular—two days in the middle of the period from the menstrual cycle onset to the day of LH peak, (2) periovulatory—the day of LH peak and the next day, (3) early luteal—2 days which are at 1/4th of the period from LH peak to the onset of the subsequent menstrual cycle, (4) mid-luteal—2 days which are at 1/2 of the period from LH peak to the onset of the subsequent menstrual cycle, and (5) late luteal—the last two days prior to the onset of the next menstrual cycle. We did not include the early follicular timepoint (i.e., onset of menses) due to the possibility of the reporting bias brought on by the menses onset, which may result in inaccurate model estimates. For descriptive purposes, we provided food craving ratings of the 4 study groups across the specified 5 timepoints, as well as the early follicular subphase, in Appendix A.

Anthropomorphic Measures. At the baseline visit, we measured waist circumference (WC) by palpating the hip area to locate the right ilium of the pelvis and positioning the tape (parallel to the floor) in a horizontal place at the level of the measurement mark [11]. For BMI, we collected the participants’ height and weight measurements using a standard measuring board and a digital scale. 

Irritability and Depression. We calculated two cardinal symptoms of PMDD from the DRSP questionnaire—depression (felt depressed, felt hopeless, felt worthless or guilty, slept more, had trouble sleeping, and felt overwhelmed) and irritability (anger and/or irritability and conflicts with people) [10,12]. Internal consistency for these measures was identified as 0.90 in a study by Endicott et al. (2006) with the intraclass correlation coefficient of 0.8 or greater [10].

High-Sensitivity C-Reactive Protein. Following standard procedure for serum separation, 1 mL of serum was frozen and sent on dry ice to ARUP laboratories, a CLIA-certified organization, for analysis. hs-crp levels were determined using quantitative immunoturbidimetry.

### 2.3. Group Categorization

We stratified the study participants into two categories: with abdominal obesity (AO) and abdominally lean (AL), according to the International Diabetes Federation [13] threshold for abdominal obesity (i.e., 80 cm or greater WC for women across ethnic backgrounds). We defined the PMDD diagnosis according to DSM-5-specified criteria [14] as prospective symptom ratings over 2 or more menstrual cycles. We identified PMDD+ participants as those with a 30% or greater increase in 5 or more symptoms, one of which had to be negative mood, relative to the range of the scale used by each woman during the week before compared with the week after menstruation [15]. Hence, the present analysis included 4 groups of participants: AO:PMDD+, AL:PMDD+, AO:PMDD−, and AL:PMDD−. Characteristics of study participants according to group are specified in Table 1. For descriptive purposes, we present premenstrual symptomatology according to the diagnosis in Appendix A. 

### 2.4. Statistical Analysis

We compared the demographic characteristics of the study groups using the Chi Square (or Fisher Exact) test for categorical, and the analysis of variance (ANOVA) for continuous variables (Table 1). Following the inverse transformation of the food craving data (outcome variable), we implemented the linear mixed-effect (LME) models with subphase, group and their interaction as predictor variables. The following were the codes menstrual cycle subphases: MF (mid-follicular), PO (periovulatory), early luteal (EL), mid-luteal (ML), and late luteal (LL). The food cravings outcome was a continuous variable, while group and subphase were factors. In this unadjusted model, the AL:PMDD− group and the MF subphase were set as references. Visual qq plot examination of standardized residuals vs. fitted values indicated a normal distribution (Appendix A). We obtained group difference estimates at each subphase using the “emmeans” function in R. In the adjusted model, we adjusted for age due to its inverse association with food cravings [16], age of menarche due to its inverse association with PMDD risk [17], and BMI to ensure that the observed effect is associated with abdominal adiposity, and not overall obesity. Hence, in the adjusted model, these measures were included as continuous variables. We assessed group*subphase*irritability and group*subphase*depression as separate predictive models of food cravings. Finally, we analyzed hs-cpr using general linear modeling (glm) (gaussian family). We modeled premenstrual food cravings as the outcome, and diagnosis by hs-cpr as the predictor. Only diagnosis (PMDD+ vs. PMDD−), and not AO, was modeled as this analysis was performed on a more limited number of 15 PMDD− and 7 PMDD+ participants. 

## 3. Results

### 3.1. Food Cravings According to Subphase and Group

Analysis of the unadjusted model revealed main effects of subphase (F = 13.48; *p* = 2.62 × 10^−10^), group (F = 4.01; *p* = 0.01), and subphase*group interaction (F = 3.28; *p* = 0.0001) (Figure 1). Relative to the reference MF subphase and AL:PMDD− group, food cravings significantly increased in the LL subphase (*t*-value =2.50; *p* ≤ 0.01), and the AL: PMDD+ group had higher food cravings in the LL subphase (*t*-value = 3.34; *p* ≤ 0.001) (Appendix A). Analysis of the adjusted model revealed that the significant effects of the unadjusted model remained such, with increased food cravings in the LL subphase (*t*-value = 2.54; *p* ≤ 0.05) and increased LL food cravings in the AO: PMDD+ group (*t*-value = 2.37; *p* ≤ 0.05). The analysis contrasting groups within each subphase showed that the AO: PMDD+ group had a highly significant increase in food cravings relative to all three study groups in the LL subphase: the AL: PMDD− group (*t*-ratio = 5.27; *p* ≤ 0.0001), the AL: PMDD+ group (*t*-ratio = 4.78; *p* ≤ 0.0001), and the AO: PMDD− group (*t*-ratio = 4.87; *p* ≤ 0.0001) (Table 2). No other contrasts were significant.

### 3.2. Food Cravings According to Group, Subphase and Affective Symptomatology

The irritability analysis showed a main effect of irritability (F = 18.92; *p* ≤ 0.001), group (F = 3.04; *p* ≤ 0.05) and irritability by group interaction (F = 2.76; *p* ≤ 0.05) on food cravings. Subphase (F = 1.42; *p* = ns), irritability*subphase (F = 0.10, *p* = ns), group*subphase (F = 1.04, *p* = ns), and irritability*group*subphase (F = 1.03; *p* = ns) were not significant. The depression analysis showed a main effect of depression (F = 39.79; *p* ≤ 0.001) and subphase (F = 2.63; *p* ≤ 0.05) on food cravings. Group (F = 0.10; *p* = ns), depression*group (F = 0.41, *p* = ns), depression*subphase (F = 2.20, *p* = ns), group*subphase (F = 0.78; *p* = ns) and depression*group*subphase (F = 0.88; *p* = ns) were not significant.

### 3.3. Association between Food Cravings and hs-crp in the LL Subphase in PMDD

hs-crp was significantly associated with food cravings (*t*-value = 2.17, *p* ≤ 0.05), and the interaction between diagnosis and hs-crp was significantly predictive of premenstrual food cravings (*t*-value = 2.30, *p* ≤ 0.05) (Figure 2). This finding indicates that the correlation between premenstrual food cravings and hs-crp was significantly different (i.e., positive) in the PMDD+ group relative to the PMDD− group.

## 4. Discussion

Food craving, which increases premenstrually [5,6], is a complex behavior regulated by intricate physiological and psychological processes that, if disturbed, may contribute to weight gain [18]. In the present study, we show that women with abdominal obesity and PMDD rate their food cravings higher in the LL subphase of the menstrual cycle compared to lean women with PMDD or women without PMDD, whether they are lean or have abdominal obesity. These differences persisted after adjusting for age, age at menarche, and BMI. Food craving ratings were not different across the study groups at any of the remaining menstrual cycle subphases. The enhanced rating of food cravings was not distinctively related to affective symptomatology in the AO:PMDD+ group relative to the other groups. Finally, hs-crp was significantly correlated with food cravings in the LL subphase in the PMDD group, which may mean that food craving results in an enhanced food intake in PMDD, and the resulting increase in this inflammatory marker. 

Several epidemiologic studies have demonstrated a bi-directional relationship between obesity and PMDD. A cross-sectional study by Masho et al. [5] found a nearly three-fold increase in risk in PMDD in women with vs without obesity, while an analysis of the prospective data from the Nurses’ Health Study 2 by Bertone-Johnson et al. [19] demonstrated that the risk for PMDD increases by 3% for every 1 kg per meter square increase in BMI. The correlation between obesity and PMDD in adult women also extends to adolescent girls [20] and non-Western populations [21]. Investigators from the Growing Up Today Study (1996–2013) [2] used prospective data to clarify the relationship between childhood BMI and subsequent risk of premenstrual disorders (premenstrual syndrome—the less severe form of PMDD, and PMDD). The results showed that baseline childhood (i.e., prior to menarche) body size is associated with an increased risk for developing PMDD in young adulthood, suggestive of shared physiological mechanisms between the two conditions. Results of the present study suggest that these mechanisms are expressed behaviorally in a temporal fashion in the late luteal subphase of the menstrual cycle. 

In addition to experiencing the core, affective symptoms of the disorder, women with PMDD self-report heightened food cravings. These symptoms are, in fact, mildly elevated in the luteal phase across reproductive-age women; however, they are more pronounced in women with PMDD. Hartlage et al. [6], for example, found that increases in self-reported appetite and food cravings in the late luteal vs. the mid-follicular subphase of the menstrual cycle were 15% and 24% in reproductive-age women, and 92% and 85% in a separate cohort of women with PMDD, respectively. Furthermore, summarizing findings from 19 studies, Buffenstein et al. [22] showed a significant energy intake increase of approximately 10% (240 kcal) on average in the luteal relative to the follicular phase in healthy reproductive-age women, and concluded that “shifts in energy expenditure during the menstrual cycle imply that energy intake should change in parallel if women are to maintain energy balance”. Indeed, resting energy expenditure (REE) significantly increases in the luteal relative to the follicular phase [4]. Hence, the luteal phase increase in food intake and craving in healthy reproductive-age women is a natural process for maintaining energy balance. 

Results of the present study show that this natural process in women with PMDD and obesity is dysregulated, and does not appear to be associated with PMDD symptomatology. Instead, it may be the result of the shared neurocircuitry between the two conditions. Considering the cyclical and menstrual cycle phase-dependent involvement of eating-related symptoms in PMDD, it is significant that many of the neurocircuit changes resulting from various aspects of appetitive food imagery processing in the orbitofrontal cortex (OFC), insula, amygdala, ventromedial prefrontal cortex (vm PFC), and striatal–pallidal neurocircuits [23,24] are also observed in PMDD in a phase specific manner [25]. For example, women with PMDD displayed a higher mid-insula activity in the luteal vs. follicular phase of the menstrual cycle on a response inhibition task—an effect opposite from the healthy controls who decreased their mid-insula activity in the luteal phase [26]. A hub for primary convergence of interoceptive signals [27], the mid-insula is reactive to food imagery; thus, the sensorimotor integration functions of this subregion may be sensitive to premenstrual changes [28] and linked to interoceptive aspects of food cravings. In fact, a recent meta-analysis [29], evaluating brain regions that exhibit disrupted activation during interoception across different psychiatric conditions, identified altered mid-insula activity as a common, transdiagnostic neural locus for detecting and regulating the body’s internal state, and a potential neural marker of psychopathology. This and other potential neurobiological overlaps between the two conditions should be carefully examined in future studies.

Abdominal obesity is related to the excess accumulation of visceral fat that significantly contributes to systemic inflammation by producing large amounts of inflammatory adipokines such as tumor necrosis factor alpha (TNFα), hs-crp, interferon gamma (IFNΎ), interleukin-6 (IL-6), interleukin 1 beta (IL1β) and many others [30]. These adipokines correlate with the volume of visceral fat and significantly predict several cardiometabolic risk parameters. They appear to correlate with premenstrual symptoms. For example, analyzing data from 2939 reproductive-age women in the longitudinal Study of Women’s Health Across the Nation (SWAN), Gold et al. [31] identified a positive association between food cravings/weight gain/bloating and hs-crp. As this analysis did not specifically evaluate premenstrual food cravings or the official PMDD DSM-5 diagnosis, and data in the literature on this topic are lacking, we preliminarily examined the relationship between premenstrual levels of hs-crp and food cravings according to group (PMDD+ vs. PMDD−), showing a significant positive correlation between the two in the PMDD+, but not the PMDD− study participants. Our analysis was underpowered to detect abdominal obesity*PMDD diagnosis interaction; nonetheless, it preliminarily shows late luteal association between hs-crp and PMDD that may predispose this segment of reproductive-age women to cardiometabolic risk. 

Food craving is a complex behavior regulated by intricate physiological and psychological processes. Its disruption contributes to excessive energy intake and weight gain [32], and women with PMDD are at risk for weight gain. We here show that the correlation between obesity and PMDD does not appear to be related to the affective symptomatology in PMDD. Nonetheless, we consider the present findings to be hypothesis-generating due to the small sample size and the self-reported nature of food cravings. The results presented here will ideally need to be confirmed using more precise methods, such as repeated sampling of dietary intake in real time and in context. Also, modern approaches, such as functional magnetic resonance imaging (fMRI), will be critical in identifying mechanisms of the observed effect. Finally, our findings would have been more robust if we controlled for any potential dieting. 

In summary, women with abdominal obesity and PMDD reported higher late luteal phase increases in food craving relative to both women with abdominal obesity and abdominally lean women without PMDD, as well as abdominally lean women with PMDD. This finding is suggestive of a biological process which predisposes this segment of population to a future cardiometabolic risk. Given the temporal nature of the observed effect in the late luteal (i.e., premenstrual) subphase of the menstrual cycle, should the present finding be replicated, targeted interventions may be developed to improve health in this at-risk population. 

## Figures and Tables

**Figure 1 nutrients-15-05000-f001:**
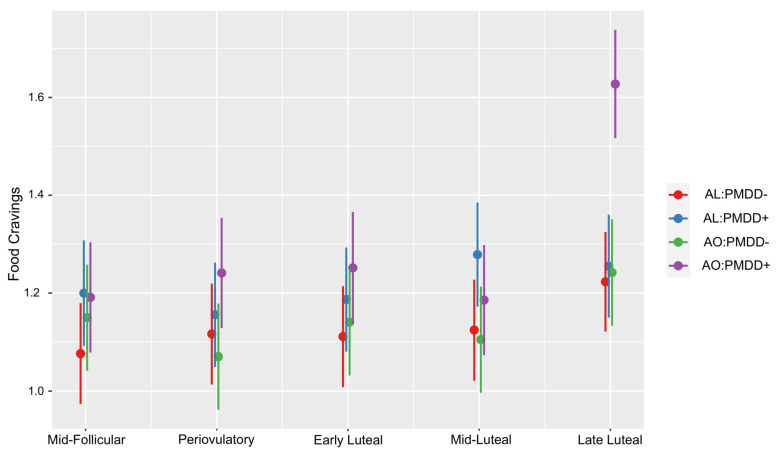
Food craving ratings from the mid-follicular to the late luteal subphase of the menstrual cycle according to the study groups. In the late luteal subphase, the AO:PMDD+ group rated food cravings higher than the remaining three groups (AO:PMDD+ with abdominal obese and PMDD; AO:PMDD− with abdominal obesity and without PMDD, AL:PMDD+ abdominally lean with PMDD; AL:PMDD− abdominally lean without PMDD).

**Figure 2 nutrients-15-05000-f002:**
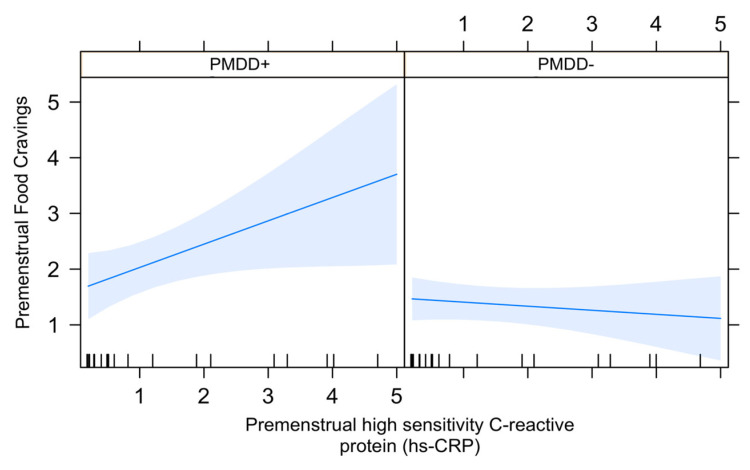
Association between premenstrual food cravings and hs-crp according to diagnosis. The association in the PMDD+, but not the PMDD− group, was statistically significant (*p* ≤ 0.05).

**Table 1 nutrients-15-05000-t001:** Demographic characteristics according to study groups.

Demographic Variable	Category	Diagnostic Category				*p* Value
		AL:PMDD− (n = 16)	AL:PMDD+ (n = 14)	AO:PMDD− (n = 15)	AO:PMDD+ (n = 13)	
Age		25.12 (4.68)	26.57 (5.48)	27.26 (5.22)	25.07 (4.53)	0.56
Race	White	6	4	7	3	0.76
	Black or African American	3	2	2	2	
	American Indian or Alaska Native	0	0	0	2	
	Asian	7	7	4	5	
	Native Hawaiian or Other Pacific Islander	0	0	0	0	
	More than one race	0	0	1	1	
	Unknown/Do not want to specify	0	1	1	0	
Ethnicity	Hispanic	1	2	4	2	0.77
	Non-Hipanic	14	11	10	11	
	Unknown/Do not want to specify	0	1	1	0	
Student Status	Yes	5	7	5	8	0.33
	No	11	7	10	5	
Marital Status	Single	15	11	13	12	0.80
	Married	1	2	2	1	
	Divorced	0	1	0	0	
	Widowed	0	0	0	0	
Income	Less than $20,000	10	6	5	6	0.51
	$20,000–$34,999	2	2	5	0	
	$35,000–$49,999	1	3	1	3	
	$50,000–$74,999	2	1	3	3	
	75,000 or more	1	2	1	1	
Age of Menarche		11.92 (1.40)	12.30 (0.94)	12.36 (1.56)	11.83 (1.52)	0.75
BMI		21.47 (1.45)	20.88 (2.32)	27.80 (3.53)	27.42 (3.98)	<0.001
Waist Circumference		71.30 (3.89)	72.37 (5.05)	89.74 (6.31)	91.63 (13.61)	<0.001

Continuous variables are summarized as means (standard deviations). Data for categorical variables are presented as Ns. AL:PMDD− = abdominally lean without PMDD; AL:PMDD+ = abdominally lean with PMDD; AO:PMDD− = with abdominal obesity and without PMDD; AO:PMDD+ = with abdominal obesity and with PMDD.

**Table 2 nutrients-15-05000-t002:** Abdominal obesity and diagnosis group contrasts according to menstrual cycle subphase.

Contrast	Estimate	SE	df	*t* Ratio	*p* Value
Subphase = Mid-Follicular					
AL:PMDD− vs. AL:PMDD+	−0.12325	0.0759	189	−1.625	0.3673
AL:PMDD− vs. AO:PMDD−	−0.07334	0.076	200	−0.965	0.7695
AL:PMDD− vs. AO:PMDD+	−0.11472	0.0776	190	−1.478	0.453
AL:PMDD+ vs. AO:PMDD−	0.04991	0.0778	194	0.642	0.9182
AL:PMDD+ vs. AO:PMDD+	0.00853	0.0794	186	0.107	0.9996
AO:PMDD− vs. AO:PMDD+	−0.04138	0.0795	196	−0.521	0.954
Subphase = Periovulatory					
AL:PMDD− vs. AL:PMDD+	−0.03927	0.0753	185	−0.521	0.9539
AL:PMDD− vs. AO:PMDD−	0.0462	0.076	200	0.608	0.9295
AL:PMDD− vs. AO:PMDD+	−0.12472	0.0776	190	−1.606	0.3775
AL:PMDD+ vs. AO:PMDD−	0.08547	0.0772	191	1.107	0.6857
AL:PMDD+ vs. AO:PMDD+	−0.08545	0.0788	182	−1.084	0.6998
AO:PMDD− vs. AO:PMDD+	−0.17093	0.0795	196	−2.151	0.141
Subphase = Early Luteal					
AL:PMDD− vs. AL:PMDD+	−0.07575	0.0753	185	−1.006	0.7463
AL:PMDD− vs. AO:PMDD−	−0.02933	0.076	200	−0.386	0.9804
AL:PMDD− vs. AO:PMDD+	−0.14036	0.0783	195	−1.792	0.2803
AL:PMDD+ vs. AO:PMDD−	0.04642	0.0772	191	0.601	0.9316
AL:PMDD+ vs. AO:PMDD+	−0.0646	0.0795	187	−0.813	0.8485
AO:PMDD− vs. AO:PMDD+	−0.11103	0.0801	201	−1.385	0.51
Subphase = Mid-Luteal					
AL:PMDD− vs. AL:PMDD+	−0.15419	0.0753	185	−2.047	0.1748
AL:PMDD− vs. AO:PMDD−	0.01936	0.076	200	0.255	0.9942
AL:PMDD− vs. AO:PMDD+	−0.06117	0.0776	190	−0.788	0.8599
AL:PMDD+ vs. AO:PMDD−	0.17354	0.0772	191	2.247	0.1144
AL:PMDD+ vs. AO:PMDD+	0.09302	0.0788	182	1.18	0.6403
AO:PMDD− vs. AO:PMDD+	−0.08052	0.0795	196	−1.013	0.742
Subphase = Late Luteal					
AL:PMDD− vs. AL:PMDD+	−0.03196	0.0746	177	−0.429	0.9735
AL:PMDD− vs. AO:PMDD−	−0.01896	0.0759	196	−0.25	0.9945
AL:PMDD− vs. AO:PMDD+	−0.40446	0.0767	182	−5.275	<0.0001
AL:PMDD+ vs. AO:PMDD−	0.013	0.077	187	0.169	0.9983
AL:PMDD+ vs. AO:PMDD+	−0.3725	0.0778	174	−4.789	<.0001
AO:PMDD− vs. AO:PMDD+	−0.3855	0.079	191	−4.878	<0.0001

## Data Availability

The data presented in this study are available on request from the corresponding author. The data are not publicly available due to preserve scientific integrity of research methodology.

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
