# Peer review of "Late Luteal Subphase Food Craving Is Enhanced in Women with Obesity and Premenstrual Dysphoric Disorder (PMDD)"

_nutrients, 2023, doi:10.3390/nu15235000_

Round 1
Reviewer 1 Report
Comments and Suggestions for Authors
Studies focusing on female-specific disorders and diseases are rare and I appreciate the authors investigating the connection between PMDD and food cravings. It was an interesting read though I feel that the authors could go into more detail in the background. I also wish that more analyses were conducted. There are also some minor comments, mainly regarding abbreviations and writing out their full version.
Title
I would suggest altering the title to specify the results of the study, for example:
Dysregulated Late Luteal Subphase Increases Food Craving in Abdominally Obese Women with Premenstrual Dysphoric Disorder 3 (PMDD)
Abstract
Minor: Add abbreviations beside abdominally lean or obese (AL) and (AO) to make it easy to understand the groups in combination with PMDD.
Write out full form of LH.
Introduction
Minor: Add full form of BMI.
Major: This is a very interesting topic, it would be great if you could elaborate the introduction. Maybe add a greater discussion on what is known about the connection between PMDD and body weight gain, other symptoms that you looked at in connection with PMDD and cravings and things that can worsen or improve symptoms. Also add in a description/discussion why you looked at CRP.
Materials and methods
Minor: In section “Anthropomorphic Measures” Add full form of WC
Write out full form of LH.
Because you state in the introduction that PMDD is associated with weight status at childhood, was weight history included assessed in this study (and not only age at menarche)?
Did you assess hunger when the participants were filling out the questionnaire’s regarding food cravings? Did it take into account if any of the participants were dieting for example? Was food craving assessed at the same time of day throughout the menstrual cycle?
Major: PMDD is associated with fluid retention which may skew regular methods of assessing body weight status. Was this considered?
Why did you measure CRP, why not cort in addition for example to look at stress and cravings? Especially seeing as obesity is associated with increased inflammation. You’re not really explaining in the intro or methods why you looked at crp. Also why didn’t you look at hormones associated with cravings and adiposity, for example estradiol, leptin, insulin.
It would be helpful to add to table 1 or add an addition table with which PMDD symptoms the women had. Were there differences in symptomology between AL and AO women?
Results
Minor: The full form of MF is not written out in full until the discussion even though it is mentioned before, both in the results section and in a table. Also LL.
Is data described in section 3.2 presented in a graph?
I would move Suppl fig3 to the results instead, that is make it Figure 2.
Does the questionnaire on food cravings give any indication to which foods are more craved, for example high fat or sugar?
I would add the reasoning for not analyzing crp data for AL and AO in both PMDD groups (+/-) earlier on (before the discussion). For the individuals included, did you control for metabolic status, as obesity is commonly associated with increased inflammation? Did you look at crp in relation to any of the other symptoms, for example depression or irritability?
Discussion
In the discussion you state:
Our analysis was underpowered to detect abdominal obesity*diagnosis interaction; nonetheless, it preliminarily shows late luteal association between hs-CRP and PMDD that may predispose this segment of reproductive age women to cardiometabolic risk.
Could you add a graph of this in the supplementary?
Supplementary
Minor: Please add a figure descriptions for suppl figures.

Comments on the Quality of English LanguageThe quality of English is great, they just need to make sure to write out abbreviations in full the first time the abbreviation is used. This is missed several times, or the abbreviation is not written out until the discussion. It makes the manuscript very frustrating to read.
Author Response
We thank the reviewers for their time and effort in reviewing our manuscript. Below, please find our responses to your comments.
REVIEWER 1
Studies focusing on female-specific disorders and diseases are rare and I appreciate the authors investigating the connection between PMDD and food cravings. It was an interesting read though I feel that the authors could go into more detail in the background. I also wish that more analyses were conducted. There are also some minor comments, mainly regarding abbreviations and writing out their full version.
Title
COMMENT: I would suggest altering the title to specify the results of the study, for example: “Dysregulated Late Luteal Subphase Increases Food Craving in Abdominally Obese Women with Premenstrual Dysphoric Disorder (PMDD).”
RESPONSE: We have updated the title of the manuscript, which now reads: “Late Luteal Subphase Food Craving Is Enhanced in Abdominally Obese Women with Premenstrual Dysphoric Disorder (PMDD).”
Abstract
COMMENT: Minor: Add abbreviations beside abdominally lean or obese (AL) and (AO) to make it easy to understand the groups in combination with PMDD.
RESPONSE: We have written out abdominally lean and abdominally obese before (AL) and (AO).
COMMENT: Write out full form of LH.
RESPONSE: We have written out the full form for LH.
Introduction
COMMENT: Minor: Add full form of BMI.
RESPONSE: We have added the full form of BMI.
COMMENT: Major: This is a very interesting topic, it would be great if you could elaborate the introduction. Maybe add a greater discussion on what is known about the connection between PMDD and body weight gain, other symptoms that you looked at in connection with PMDD and cravings and things that can worsen or improve symptoms. Also add in a description/discussion why you looked at CRP.
RESPONSE: We thank the reviewer for the thoughtful response. We added the following regarding CRP, and kept the Introduction concise: “We deemed this analysis necessary as the inflammatory protein is a marker of poor dietary intake, and it is unknown whether premenstrual crp is associated with PMDD.” We further extended the Introduction in response to the Reviewer’s comments.
Materials and methods
COMMENT: Minor: In section “Anthropomorphic Measures” Add full form of WC.
RESPONSE: In section "Anthropomorphic Measures” we added full form for WC.
COMMENT: Write out full form of LH.
RESPONSE: We wrote out full form of LH.
COMMENT: Because you state in the introduction that PMDD is associated with weight status at childhood, was weight history included assessed in this study (and not only age at menarche)?
RESPONSE: We did not measure childhood weight status in the present study, though we plan to include this in our future work.
COMMENT: Did you assess hunger when the participants were filling out the questionnaire’s regarding food cravings? Did it take into account if any of the participants were dieting for example? Was food craving assessed at the same time of day throughout the menstrual cycle?
RESPONSE: The Daily Record of Severity of Symptoms was filled out between 8 PM and midnight every day; assessing each symptom over the entire day. We added this important point in the revision. The questionnaire does not assess the current hunger status. In the limitation section, we added that we did not assess dieting.
COMMENT: Major: PMDD is associated with fluid retention which may skew regular methods of assessing body weight status. Was this considered.
RESPONSE: Fluid retention is observed across reproductive age women. We, hence, did not assess this physical phenomenon as it is considered to be general.
COMMENT: Why did you measure CRP, why not cort in addition for example to look at stress and cravings? Especially seeing as obesity is associated with increased inflammation. You’re not really explaining in the intro or methods why you looked at crp. Also why didn’t you look at hormones associated with cravings and adiposity, for example estradiol, leptin, insulin.
RESPONSE: As part of a different manuscript, we assessed cortisol across the menstrual cycle, and did not see an increase in the premenstruum. As food cravings increase, but cortisol does not, we did not deem this analysis necessary. We stated in the Introduction our rationale for analyzing CRP. Evaluating estradiol, leptin, insulin are interesting topics that we may include in future analyses.
COMMENT: It would be helpful to add to table 1 or add an addition table with which PMDD symptoms the women had. Were there differences in symptomology between AL and AO women?
RESPONSE: In response to the Reviewer’s request, we analyzed additional data and added a figure which shows premenstrual symptomatology in women with PMDD vs. healthy controls. In the revised version, this is now depicted as Supplementary Figure 3. There were no differences in symptomatology between AL and AO women.
Results
COMMENT: Minor: The full form of MF is not written out in full until the discussion even though it is mentioned before, both in the results section and in a table. Also LL.
RESPONSE: We thank the reviewer for this comment. In the revised manuscript, all the subphases are defined in the statistical analysis section. Please see the following sentence:” The following were the menstrual cycle subphases: MF (mid-follicular), PO (periovulatory), early luteal (EL), mid-luteal (ML), and late luteal (LL).”
COMMENT: Is data described in section 3.2 presented in a graph?
RESPONSE: No, there is no graph. These results are described in the verbal form. If the Reviewer advises that we include a graph, please let us know and we will do so.
COMMENT: I would move Suppl fig3 to the results instead, that is make it Figure 2.
RESPONSE: In the revised version, we moved Supplementary Figure 3 to Figure 2.
COMMENT: Does the questionnaire on food cravings give any indication to which foods are more craved, for example high fat or sugar?
RESPONSE: The DRSP does not inquire about the food type. This is something we plan to assess in the future.
COMENT: I would add the reasoning for not analyzing crp data for AL and AO in both PMDD groups (+/-) earlier on (before the discussion). For the individuals included, did you control for metabolic status, as obesity is commonly associated with increased inflammation? Did you look at crp in relation to any of the other symptoms, for example depression or irritability?
RESPONSE: The reasoning for not analyzing CRP data in all 4 groups is included in the Statistical Analysis section: “We modeled premenstrual food cravings as the outcome, and diagnosis by hs-CRP as the predictor. Only diagnosis (PMDD+ vs. PMDD-), and not AO, was modeled as this analysis was performed on a more limited number of 15 PMDD- and 7 PMDD+ participants.” We did not control for metabolic status as the crp analysis is for PMDD, and obesity status did not differ between the 2 PMDD groups. CRP was not related to depression or irritability.
Discussion
COMMENT: In the discussion you state: Our analysis was underpowered to detect abdominal obesity*diagnosis interaction; nonetheless, it preliminarily shows late luteal association between hs-CRP and PMDD that may predispose this segment of reproductive age women to cardiometabolic risk. Could you add a graph of this in the supplementary?
RESPONSE: This graph was Supplementary Figure 3, that is now Figure 2 per the Reviewer’s request.
Supplementary
COMMENT: Minor: Please add a figure descriptions for suppl figures.
RESPONSE: We have added figure descriptions for all the supplementary figures.
FINAL RESPONSE TO THE REVIEWER: We thank you again for your time and effort in reviewing our manuscript.
Reviewer 2 Report
Comments and Suggestions for Authors
Is DRSP a fully validated measure for food craving? What are the psychometric properties of this measure? How do you interpret for example a difference of 2 sores on DRSP? What is a meaningful difference?
Could you provide more details on the PMDD diagnosis? This is an important grouping variable, and I wonder how PMDD is corrected with food craving, especially you included PMDD as a covariate and food craving as the dependent variable in the regression model.
My main concern is the sample size is limited and thus multivariate regression analysis is not powered to delineate the association between AO and food craving. The PMDD variable itself seems to be related to food craving too, and thus, the regression model can be problematic. I think the current data supports more hypothesis generation and I recommend you present the data in a descriptive manner. Future studies are needed to validate the hypothesis.
Author Response
We thank the reviewers for their time and effort in reviewing our manuscript. Below, please find our responses to your comments.
REVIEWER #2
COMMENT: Is DRSP a fully validated measure for food craving? What are the psychometric properties of this measure? How do you interpret for example a difference of 2 sores on DRSP? What is the meaningful difference?
RESPONSE: The DRSP is a validated measure of premenstrual symptomatology, and it includes an item assessing daily food craving. The test-retest reliability and internal consistency are high as described in Endicott (PMID: 16172836). DRSP scores are evaluated longitudinally with respect to an increase in the late luteal subphase of the menstrual cycle. For example, an elevation of a score by 0.3 on a specific item in the late luteal subphase (the period in the menstrual cycle when symptoms typically exacerbate) relative to the mid-follicular subphase (when symptoms subside) is interpreted as a 30% elevation in symptomatology, and this is considered clinically meaningful.
COMMENT: Can you provide more details on the PMDD diagnosis? This is an important grouping variable, and I wonder how PMDD is corrected with food craving, especially you included PMDD as a covariate and food craving as the dependent variable in the regression model.
RESPONSE: We thank the reviewer for pointing out this important aspect. PMDD is diagnosed when 5 or more (of the 21 total DRSP items) are elevated by 30% or higher in the late luteal subphase of the menstrual cycle (please see our description above). At least one of these 5 items must be affective in nature (7 out of 21 items assess affect), and one of the 21 items assesses food cravings. Women who were diagnosed with PMDD generally score 30% or higher on more than 5 required items; hence, although food cravings is one of the 21 items, it is not crucial for diagnosis. We included a new figure (Supplementary Figure 3) which shows elevated food cravings in women with and without PMDD, and this is, in fact, expected, as food cravings naturally increase due to the physiological state of a higher metabolic rate related to the endometrial thickening and buildup.
COMMENT: My main concern is the sample size is limited and thus multivariate regression analysis is not powered to delineate the association between AO and food craving. The PMDD variable itself seems to be related to food craving too, and thus, the regression model can be problematic. I think the current data supports more hypothesis generation and I recommend you present the data in a descriptive manner. Future studies are needed to validate the hypothesis.
RESPONSE: We have added more language regarding this point in the limitation section.
Round 2
Reviewer 1 Report
Comments and Suggestions for Authors
The authors have addressed each one of my comments and I am satisfied with all of their changes.